# It Is Not Only Data—Freshwater Invertebrates Misused in Biological Monitoring

**DOI:** 10.3390/ani13162570

**Published:** 2023-08-09

**Authors:** Paweł Koperski

**Affiliations:** Institute of Functional Biology and Ecology, Faculty of Biology, University of Warsaw, Żwirki i Wigury 101, 02-089 Warszawa, Poland; p.t.koperski@uw.edu.pl

**Keywords:** biological assessment, animal ethics, methods, mortality, nature protection, welfare

## Abstract

**Simple Summary:**

The article explains why using free-living invertebrates in biomonitoring aimed at assessing the ecological status of rivers, lakes, streams and ponds can be considered misuse. Invertebrates are excluded from ethical considerations in environmental procedures, resulting in the killing of many more individuals than necessary during such activities. Biomonitoring used as a routine method of environmental protection causes cruel deaths of up to millions of aquatic animals every year. Improperly planned procedures which result in excessive mortality have or may have a negative impact on the environment and biodiversity. The life of aquatic invertebrates, although they should be considered sensitive beings, is reduced to an informative function; they become only data useful for biomonitoring purposes. Some new methods, modifications and improvements of biomonitoring procedures that can significantly reduce freshwater invertebrate mortality are presented in the section “Future Directions”. Especially the development of effective, precise and reliable methods of survival, e.g., based on the analysis of DNA taken directly from the environment (eDNA), seems to be not only a breakthrough in biomonitoring, but also an important step towards a significant improvement in the welfare of aquatic invertebrates.

**Abstract:**

The article presents and discusses the issues of the use of free-living invertebrates to assess the ecological status of freshwater environments with different methods of biological monitoring. Invertebrates are excluded from ethical consideration in the procedures of environmental protection, which results in the killing of many more individuals during sampling than necessary. Biomonitoring is used as a routine method for environmental protection that results in the cruel death of even millions of aquatic animals annually. In many cases, the mortality of animals used in such types of activities has been shown as excessive, e.g., because the vast majority die due to unnecessary subsampling procedures. Improperly planned and conducted procedures which result in excessive mortality have or may have a negative impact on the environment and biodiversity. Their existence as sensitive beings is reduced to an information function; they become only data useful for biomonitoring purposes. The main problem when trying to determine the mortality of invertebrates due to biomonitoring activities and its impact on natural populations seems to be the lack of access to raw data presenting how many animals were killed during sampling.

## 1. Introduction

“It is clear that we have direct ethical obligations to sentient animals; but it is not at all clear that we have direct ethical obligations to entities such as species, or to biological diversity. The burden of proof should thus be on conservationists to show how killing the first to preserve the second can possibly be acceptable from an ethical point of view” [1]. Biological monitoring (biomonitoring), in the broadest sense, can be defined as activities aimed at assessing the condition of the environment, performed according to an agreed methodology and using bioindicators. It consists of the assessment of different parameters and detecting ongoing changes in ecosystems and components of biological diversity, including types of natural habitats, populations and species, as well as to assess the effectiveness of the nature protection methods used. In this approach, biological monitoring aims not only to assess the state of the natural environment, organisms and ecosystems, but also other environmental parameters, such as the degree of habitat transformation, and soil, air and water pollution. Some methods are based on the presence or relative abundance of specific taxonomic groups in the environment. The assumption of some of the biomonitoring methods, especially those aimed at assessing the presence of xenobiotics in the environment and their effects, depends on the collection of whole organisms, body fragments or various physiological secretions for analysis.

The Convention on Biological Diversity [2] encourages states that are parties to it to monitor the elements of biological diversity, with particular emphasis on its most endangered components and those representing the greatest potential value for sustainable use. This should include, in particular, monitoring the effects of processes and activities that have or may have a significant negative impact on the conservation and sustainable use of biological diversity. According to the convention, environmental monitoring should cover all levels of biodiversity from ecosystems, through to species level, to genetic diversity.

Since the 1980s, this type of activity has been most intensively carried out in freshwater environments. In European Union countries, in accordance with the requirements of the Water Framework Directive of the European Parliament (WFD) [3], the basis for this type of assessment is the analysis of the composition of the so-called “biological elements”. This term means groups of organisms with a confirmed high indicative value. Indicator organisms can be used for biological monitoring purposes at various organizational levels, from the sub-organismal level (e.g., genes, cells, tissues) through the organismal level, to populations, communities or even a whole ecosystem level [4]. Among the five biological elements used to assess the ecological status of European freshwater environments: fish, macrophytes, phytoplankton, periphyton algae and macroinvertebrates, the latter is the most often and most commonly used. Macroinvertebrates remain on the sieve with a mesh size of 0.2–0.5 mm, i.e., in practice have a body length exceeding 1 mm. It is estimated that about two-thirds of the methods for assessing the quality of flowing waters are based on benthic macroinvertebrates.

About 15,000 species of Metazoa occur in European waters, of which about 10,500 can be termed macroinvertebrates. The vast majority of those used in monitoring programs are insects, in particular their larval forms. However, the less numerous groups include water-mites (Acari: Arachnida), snails and mussels (Mollusca), leeches and oligochaete worms (Clitellata), malacostracan and other groups of so-called Crustacea. Insects contain the taxa with the highest indicative value e.g., the EPT group (Ephemeroptera, Plecoptera, Trichoptera larvae). Many species of freshwater invertebrates in Europe are endangered [5] and are particularly sensitive to habitat change as flow alterations, habitat fragmentation, long-lasting drought and pollution are their main threats [6]. The recently intensively studied global decrease in the number and biomass of insects, including ecologically specialized rare species of aquatic insects, is undoubtedly related to the degradation of freshwater environments [7].

A significant part of the waters located in areas of intensive human impact are degraded, and their functional, hydrological and chemical parameters as well as the taxonomic composition of the organisms inhabiting them significantly differ from the pristine conditions. Routine assessment of the ecological status of these environments is carried out at designated sites by official institutions appointed by the authorities to collect data on ecosystem degradation. Freshwater environments—lakes, ponds, wetlands, streams and rivers—are some of the most threatened habitat types on Earth. They contain less than three percent of the volume of water stored on Earth, but about ten percent of animal species live in them. Typically, a large part of terrestrial biodiversity is concentrated near freshwater [8]. Declining numbers of aquatic invertebrates have negative effects on ecosystems because the populations of many species are essential for their function. These organisms filter a huge amount of edible particles suspended in water, control prey populations as predators, constitute the food base of fish species and consume periphytic algae and dead organic matter, which accelerates the biomass turnover. The activity of certain groups reduces the effects of eutrophication and intensifies the self-purification processes in watercourses. Huge swarms of winged mayflies, true-flies and caddisflies after their emergence constitute an irreplaceable food base for many terrestrial vertebrates [9]. As significant reductions in their abundance may cause disturbances in the functioning of the ecosystem, it means that excessive mortality can be regarded as a kind of environmental degradation.

How should we understand the statement that invertebrates are ‘misused’ in the title of the article and what is this ‘misuse’ supposed to consist of? In our opinion, this misuse means, above all, excessive suffering and excessive mortality as a result of biomonitoring procedures that affect a large number of aquatic animals. Most often it takes the form of non-selective bulk sampling with the use of sieves, dip-nets, various samplers and toxic substances with a preservative effect. As a result of such activities, the number of animals sampled and killed exceeds, sometimes many times, the numbers necessary to properly apply the ecological status assessment protocols. Researchers often kill far more animals in each sampling effort than necessary to conduct a proper assessment of the ecological status of the habitat for fear of collecting too little data. Reducing suffering and reducing mortality is generally not a primary or secondary objective underlying any official assessment methods. The term ‘excessive mortality’ should be considered in detail. From an ethics point of view when treating animals as sentient beings (animal welfare), ‘excessive’ can be generally referred to any number of beings that we endow with inherent moral value, without attempting to prevent it. From the point of view of nature conservation, the meaning of this term is more difficult to define. It certainly concerns a situation where the level of animal mortality permanently threatens the stability of the population. In a situation such as that described in this paper, however, it is impossible to even attempt to assess the strength of such an influence. So, it seems justified to use the term ‘excessive mortality’ to describe the effects of sampling methods in which many thousands of specimens are killed non-selectively, most of which are not used later for any purpose [10].

### Use of Freshwater Invertebrates in Biomonitoring

A typical method of assessing the ecological status of a freshwater environment, in accordance with the requirements of the WFD, is based on the following points: (i) selection of the field site, as assigned to the appropriate abiotic type; (ii) quantitative sampling of invertebrates from the appropriate bottom area, using specialized equipment; (iii) preserving the samples using a suitable substance; (iv) random sampling in the laboratory from the preserved organisms of sub-samples of the total sample until the appropriate number of specimens are obtained, e.g., the minimum for proper evaluation; (v) identification of animals selected in sub-samples to the required level (typically the level of family); (vi) calculation of the value of the multimetric biotic index on the basis of taxonomic composition and richness of the invertebrate fauna; (vii) determination of the class of the ecological status (on a five-point scale) on the basis of the final index score depending on the habitat type [11]. A detailed description of the protocol is available online [12].

It should be noted that new versions of protocols developed for the assessment of freshwater environments increasingly contain propositions intended to reduce the mortality of at least some invertebrates, without decreasing the quality of the assessment. There, you can find suggestions to return to the environment as many individuals as possible whose identification is possible already in the field, e.g., mussels from the Unionidae family, crayfish, as well as to review samples in the field for the presence of protected species and if they occur, their presence should be recorded in field protocol, and the animals removed from the sample and left in the environment [12]. In practice, during field analysis, such a procedure may be only partially effective, e.g., due to the inability to see the younger (smaller) developmental stages of large animals and the inability to distinguish in situ individuals of protected and unprotected species (e.g., dragonflies or beetles).

The methods of biological assessments of freshwater environments differ in both the sampling methods and the biological variables used [13]. Most published studies and reports for different reasons do not mention the total number of animals killed (preserved) but only the number of animals used for analysis. Therefore, it is very difficult to directly compare the levels of invertebrate mortality during routine biological monitoring carried out by different methods. Important factors affecting such mortality rates are: sampling area, methods, intensity, mesh size, sample replications and handling methods of trapped animals (live sorting, preservation of the whole sample, non-lethal collection or identification of selected taxa). Some procedures which are applied in the laboratory post-preservation (i.e., the use of magnification during sorting, sub-sampling and the level of identification) appear to have no effect on mortality, because they only reduce the number of animals used for analysis after preservation, and thus killing all; regardless of the method, it is never an easy “humanitarian” death [11].The effects of using the described sampling methodology in the routine monitoring of flowing waters in Poland between 2011 and 2018 were analyzed in detail [10,11]—this will be used to illustrate general patterns. The methods of biomonitoring in lakes, dam reservoirs and coastal brackish waters used in Poland are so similar that the effects caused by them are probably similar [12]. It can be assumed that the conclusions from a detailed analysis of the effects of sampling in routine monitoring in other countries using methods based on similar assumptions and defined by the requirements of the WFD would be similar, e.g., [14,15]. From the point of view of the considerations discussed in this article, the most important direct effects of the described procedure of preserving entire bulk samples and the use of subsample analyses in the monitoring of flowing waters are:The number of aquatic invertebrates killed (preserved) during sampling is sometimes very high, which significantly exceeds the minimum number necessary for a correct assessment of the state of the environment;A significant number of the animals killed during conservation are not used for analysis, which receives much less abundant sub-samples;There are significant variations in the numbers of animals killed during maintenance, involving the taking of samples for analysis by staff from different laboratories, following the same protocols and often in very similar types of environments.

The number of animals killed during the sampling procedure exceeds, on average, 12 times the number necessary for proper analysis (the described method requires placing at least 350 identified individuals in the database intended for the calculation of the index). In sixty-one percent of the samples, at least five times as many animals were killed than the number sufficient for analysis, in some cases reaching even over 200–500 times more. Moreover, it can be estimated that these numbers are actually even higher due to the fact that for taxa occurring in high densities, the procedure allowed to record only the first 100 individuals sampled, but typically it is not noted in the archived data [12].

As a result of the described subsampling procedure, 80.4% of the animals killed as a result of conservation were not used for the assessment of the environment and died, so to speak. Among the animals of which this type of misuse applied, insects (63.4% of individuals) and Malacostraca (17.3%) predominate. Identifying animals only at the family level means that the collected data cannot be used for scientific purposes (e.g., biodiversity analysis) and it also makes it impossible to reliably assess the impact of the procedure on the populations of rare and protected species. For example, over 10,000 individuals of the mussel family Unionidae were killed, among which there are rare, vanishing and protected species, of which about 8300 individuals, despite being killed, were not included in the analysis. In the case of the Heptageniidae family (Ephemeroptera, mayflies), which also includes rare, endangered and protected species, these numbers are 125,455 (killed specimens), 24,577 (number enough for analysis) and 100,878 (not used for analysis), respectively. In total, there were 2,817,777 specimens which belonged to families, including species covered by various forms of legal protection.

It has been estimated that between 2012 and 2019, at least 12.7 million invertebrates (more than 8.6 million aquatic insects) fell victim to biomonitoring in Poland’s watercourses (Figure 1), and according to very rough estimates, on a European scale it was about 18 million animals per year [10,11]. It is very difficult to assess what effects such mortality has on aquatic invertebrate populations and the functioning of ecosystems. This type of impact can only be expected locally in specific environments or directed in isolated populations living in low densities. The differences in the average numbers of invertebrates killed between laboratories performing routine biomonitoring can be extreme (up to 35 times). Importantly, this does not usually stem from the effect of differences in animal densities, resulting from the ecological and geological differences between environments. The average number of invertebrates collected by the staff of different laboratories from one square meter patches at the bottom of watercourses of the same abiotic type (and thus very similar in terms of hydrological, geological and ecological parameters) differed even up to thirty-two times. In the analysis, much attention was paid to the hypothetical reasons for such large differences in mortality values during sampling [10]. The overall variance associated with the total number of insects killed during sampling is explained mainly by: (i) the specificity of the laboratory and (ii) the specificity of the abiotic type; the first factor is much more important than the second one. The remaining analyzed factors are of very little importance. This clearly indicates that it is the “human factor” that is difficult to define—the level of empathy in dealing with invertebrates, which varies greatly among individual people taking samples, is the most important in this case. More important than ecological factors, freedom in the use of the sampling equipment allowed by the procedure and even the aforementioned lack of mechanisms limit the maximum sample abundance.

## 2. Discussion

Freshwater invertebrates are still treated as non-sentient, thus are devoid of intrinsic value and therefore do not require any form of ethical protection. Deep differences in the endowment of vertebrates and invertebrates with moral entitlement are still clearly visible and take the form of a clear asymmetry [16]. Ethical conflicts are inevitable as a result of large-scale biomonitoring activities resulting in the mass non-selective killing of invertebrates. It seems that the main reason is the simultaneous classification of the animals used into various, often inseparable, functional categories, to which moral rights are granted (or not) to varying degrees. The simplified effects of such classification are shown in Figure 2. Recognizing a given group of animals only as “elements of the population with high indicative value in biomonitoring procedures” automatically excludes them from the sphere of ethical protection and in practice reduces their existence only to an information function—their existence as sensitive beings and all of their life’s interests from that point on become just data for the purposes of biomonitoring. In other words, it deprives them of an intrinsic ethical value, giving them only an instrumental value, significant only from the point of view of human benefit. Removing invertebrates from the sphere of ethical protection makes their killing during monitoring an ethically neutral activity, regardless of the number of animals killed. It should be emphasized that such commonly held beliefs about relations between humans and animals are usually based on a personal worldview shaped by religious principles, as argued, e.g., [17,18]. Probably for some people participating in this type of activity, obtaining data by counting the bodies of killed animals does not differ, in ethical terms, from the assessment of the concentration of microorganisms or plant biomass in water, or even the concentration of chemical compounds dissolved in water [11]; it then becomes an act perceived positively.

In the classic twentieth-century publications focused on animal ethics, questions of moral duty are limited to homoiothermic vertebrates. It is necessary to pay attention to the real revolution of ideas that we have been dealing with in the last twenty years, with the results of studies in the fields of neuroscience and behavioral biology clearly demonstrating the presence of advanced cognitive functions, individual behavioral types and emotions in invertebrates. According to them, empirical evidence unequivocally commands abandoning the habit of treating invertebrates as primitive beings whose behavior is based solely on simple reactions to stimuli and automated instincts [19,20], and unequivocally requires perceiving them as sentient. The relatively universal agreement in the world of science of ethical and legal protection so far prevails to cephalopods and decapods [21].

The concept of evolutionary inclusive ethics [22], postulating the inclusion of invertebrates in the sphere of ethical protection on the basis of recognition of their high degree of sensitivity and the presence of advanced cognitive abilities, is met with a particular response among specialists in animal ethics. The authors argue that the near-total exclusion of invertebrates from ethics-based science policy is not due to the current state of scientific knowledge, but is mainly due to: (i) a naive reading of evolutionary theory that invertebrates are a lower category than vertebrates; (ii) the a priori and false assumption that small brains cannot provide advanced ways of knowing or feeling; (iii) human biases and cognitive-affective biases (e.g., feelings of disgust) that distort moral judgments. The difficulties in initiating practical changes regarding the welfare of invertebrates are summed up by Mather in the introductory article of this volume [17], where she writes that the traditional anthropocentric approach to invertebrates is based on: “our lack of knowledge, our negative attitudes and our misunderstanding of their cognitive abilities..”. There is both a logical and scientific basis for including at least some invertebrates in considerations of ethical eligibility. In particular, consistency in the moral treatment of non-human animals is essential: the same characteristics, criteria and reasoning that justify moral protection for vertebrates should serve to extend similar protection to at least some invertebrates. This concept is widely commented on in the world of specialists in animal rights and animal minds [18,20] and it seems to also apply to at least some aquatic invertebrates used in biomonitoring. It seems that an effective way to resolve the ethical conflicts that arise during the implementation of biomonitoring procedures may be the application of the Animal Sentience Precautionary Principle [23]. According to it, doubts about the existence of sentience in a given animal should not lead the researcher to limit the ways of avoiding harm to him. The author proposes to assume the existence of the ability to feel in all representatives of species closely related to the animal, for which it was confirmed experimentally—this also applies to many species of freshwater insects and crustaceans used in biomonitoring.

Some ethicists, however, question the sense of using criteria based on comparing the cognitive abilities of different animals when granting moral rights, treating it as a manifestation of “neo-speciesism” [24]. According to such concepts, it is the welfare of each individual, and not “ingenuity” and relative similarity to human behavior, which should be the basis for moral protection [25].

## 3. Conclusions

I.Freshwater invertebrates used in the most common biomonitoring methods are still treated as non-sentient beings, thus are devoid of intrinsic value and therefore do not require any form of ethical protection.II.This leads to misuse, the most common of which is excessive suffering and excessive mortality during sampling. Most often, it takes the forms of non-selective bulk sampling and the use of sub-sampling procedures with the result that the number of animals killed exceeds, sometimes many times, the numbers necessary to properly apply the ecological status assessment protocols.III.The parallel classification of freshwater animals used in biomonitoring into different categories with different ethical status leads to inevitable ethical conflicts (see Figure 2).IV.It is difficult to assess the ecological effects of such treatment of free-living populations of freshwater invertebrates, but many individuals of protected species also fall victim to such biomonitoring.V.The search for improvements and new methods of biomonitoring seems to be necessary and obvious for many specialists. Especially, the development of efficient, precise and reliable methods based on environmental DNA (e-DNA) analysis seems to be not only a breakthrough in biomonitoring, but also an important step towards a significant improvement in the welfare of aquatic invertebrate.

## 4. Future Directions

The need for change in the approach to the use of invertebrates in biomonitoring seems obvious, both on the part of scientists [26] and practitioners [11]. The discussed problem is a model example of a fundamental conflict between environmental ethics and animal ethics [27]. It seems to be a difficult problem to demonstrate, in accordance with ethics, that the death and suffering of many free-living animals, for which there are logical and scientific premises that they are sentient, can be justified by maintaining the species and ecosystems in a good ecological condition [1]. The benefits of biological monitoring are enormous and undeniable, but raise ethical conflicts when used as a justification for this type of abuse. It seems likely that the lack of generally accepted ethical standards for killing insects may affect personal attitudes related to ethical sensitivity and thus individual behavior [28]. The priorities underlying modern monitoring procedures appear now to be obsolete because they do not take into account the ethical aspect of killing sentient individuals. Large scientific projects in which terrestrial insects are monitored for biodiversity assessment also result in significant mortality [29]—tens of millions specimens). However, they differ from the results discussed in this paper by a very important feature. Namely, unnecessary mortality was reduced there to a minimum—all individuals are by definition identified with maximum accuracy by specialists or archived for future identification. The goal is to know biodiversity, so regardless of the number of individuals killed, the death of each of them matters. In the case of the discussed data on the ecological quality assessment, the majority of individuals are intentionally, but unnecessarily, killed and are later not used for anything. This kind of ethical insensibility is also in contradiction with the more and more commonly formulated proposals to grant ethical value not only to individuals but also to biodiversity, as such [30]. An increasing amount of evidence from experimental research is revealing that many groups of invertebrates have complex behavior and advanced mental potential: substantial perceptual ability, feelings and emotions, pain perception, long- and short-term memory [31], learning abilities, cognitive perception and individual differentiation of behavioral types [32,33]. The view that any activity that causes pain or death in sentient animals should be limited and justified only if the benefits to other organisms or the ecosystem as a whole are significant is becoming more and more accepted [20].

Reducing the negative impact of scientific research, including environmental research, on invertebrates has been postulated for years from ethical positions. Therefore, alternatives to the lethal methods of biomonitoring widely used so far are also sought. Non-lethal methods based on the composition of macroinvertebrates are very few and consist of the morphological identification of digitally recorded images of animals [34,35,36]. Projects of the photographic identification of various groups of terrestrial and marine invertebrates are being tested, e.g., [37]. For many years, methods have been developed to assess the impact of various anthropogenic stressors on freshwater ecosystems based on the analysis of biomarkers (stress and enzymatic responses, endocrine disruptors, trophic tracers, energy metabolites, genotoxic indicators, histopathological and behavioral alterations and genetic markers) from various groups of invertebrates [38].

The easiest way to avoid most of the described ethical conflicts is a complete resignation from the use of animals in biomonitoring and limiting it to the analysis of non-sentient organisms only: microorganisms (bacteria, algae, protists) and aquatic plants [11]. A reduction in mortality during traditional, routinely used methods of the biomonitoring of freshwater environments can be achieved by the following modifications:1.Pre-sampling. Before the actual biological sample is taken, a preliminary sample should be taken near the test site, e.g., twenty-five percent of the appropriate sample, in order to estimate the density of invertebrates. Depending on the results of this initial count, the appropriate sample required can be estimated, covering only an area adequate to obtain the final number of animals needed for the assessment. Only those animals will be killed and preserved. This procedure can be simplified by the use of photographs and videos along with image analysis software (using self-learning systems based on artificial neural networks) that automatically recognizes and counts objects.2.A similar procedure can be carried out by using modular artificial substrates placed at the sampling point several weeks in advance [32]. Depending on the preliminary assessment, the fauna would be collected from an appropriate number of modules to obtain the final number of animals in the sample close to the minimum required.3.The application of numerical analysis procedures such as rarefaction, and their use to create empirical saturation curves, will allow the estimation of the maximum abundance of macrobenthos in a sample necessary for a reliable estimate of taxonomic richness [10].

Non-lethal methods used to collect animal body fragments, their tissues or gametes are usually based on procedures that cause stress and suffering in the tested animals [39,40]. They were developed to reduce the mortality, suffering and discomfort of the analyzed individuals in comparison with the methods used so far, therefore they should be considered as activities conducive to animal welfare; however, the ethical assessment of such activities should be approached with great caution. The term ‘welfare’ used to refer to free-living animals is ambiguous. Especially when used in relation to invertebrates used in environmental studies—it may in fact mean [41] simply a reduction in suffering during exploitation or killing and does not take into account any basic vital interests of animals.

Methods using environmental DNA (e-DNA) are probably the most promising and fastest growing alternative, but it should be emphasized that they were not created for ethical reasons and not all of them can be considered “non-lethal”. The main motivation for their development was, in fact, the possibility of a quick and effective assessment of the taxonomic composition of organisms in the environment without the need for time-consuming and labor-intensive morphological identification based on expert knowledge. E-DNA can broadly be defined as the genomic DNA of multiple organisms obtained directly from the environment. This can include approaches that focus on extracting the DNA of taxa presently residing within the sampled matrix, or the collection of degraded fragments of DNA that persist within the environment [42]. Specialists emphasize very rapid development and great prospects for methods based on e-DNA [43], and predict a huge increase in their importance for the purposes of biomonitoring [44]. The effectiveness of e-DNA-based methods in determining the taxonomic composition of organisms in monitoring is still widely discussed. Enthusiastic opinions announcing the rapid replacement of traditional methods [45,46] must face counterarguments pointing to their serious limitations and low reliability [47,48]. Various biological and technical limitations still impede the implementation of the environmental genomics for routine monitoring applications. These limitations mainly stem from the fact that the methods sample fundamentally different units of presence (molecules vs. individuals), resulting in different biases affecting richness, abundance and taxonomic composition [41,49]. The richness of “molecular species” should not be considered analogous to species richness [44,50]. The biggest problem associated with the use of such methods in routine biomonitoring still seems to be the reliable reconstruction of the number or at least the relative abundance of individual taxa in the environment based on DNA fragments found in an environmental sample [51,52]. Certainly, the development of methods based on e-DNA analysis leading to an effective and reliable reading of the quantitative taxonomic composition of aquatic invertebrates with appropriate taxonomic resolution will not only be a breakthrough in biomonitoring, but also an important step towards a significant improvement in the welfare of aquatic invertebrates.

## Figures and Tables

**Figure 1 animals-13-02570-f001:**
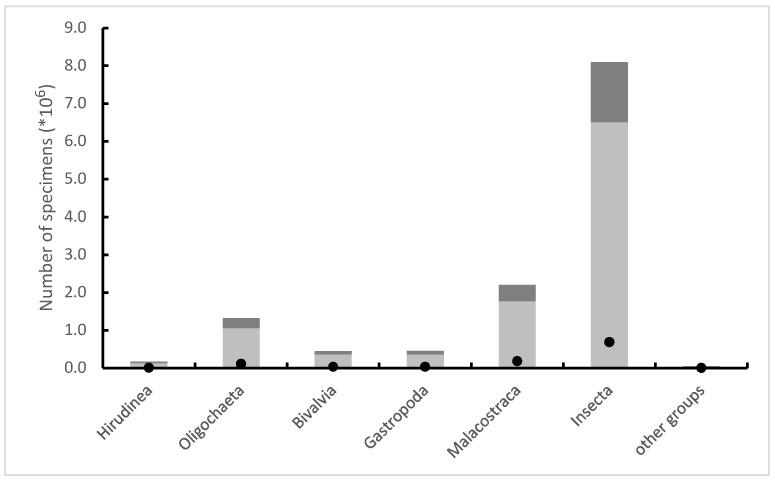
Estimated number of animals from the most important taxonomic groups that were killed during sampling for the purposes of biomonitoring in Polish watercourses in 2012–2019, with the number of specimens used for the analysis (dark grey) and the number of individuals necessary for the correct analysis (black dots) shown.

**Figure 2 animals-13-02570-f002:**
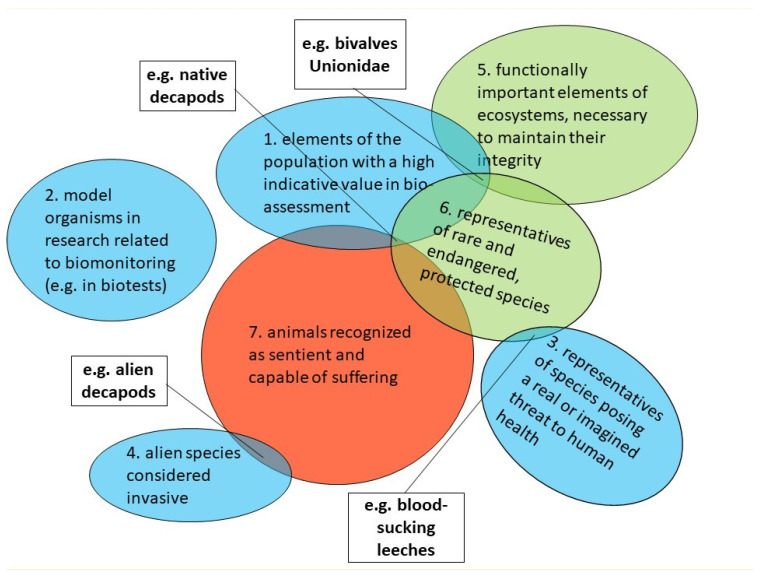
Freshwater invertebrates used in biomonitoring assigned to different functional categories that differ in the moral entitlements granted to them by humans (blue figures—no moral entitlements, green—some moral entitlements granted on the basis of environmental ethics (as exemplars of species), red—moral entitlements granted on the basis of animal ethics (as sentient beings). Ethical conflicts arise where figures of different colors overlap—examples of such animals are shown in empty rectangles.

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
