# Peer review of "It Is Not Only Data—Freshwater Invertebrates Misused in Biological Monitoring"

_animals, 2023, doi:10.3390/ani13162570_

Round 1
Reviewer 1 Report
The manuscript deals with a very important subject, often overlooked by the researchers in the struggle to get good, quality monitoring or bioassessment results. It is important to discuss about the future of the water quality assessment methods in this sense - not to jeopardize biodiversity in the attempt to save it.
I would appreciate more references, especially since this is a review paper.
Repeating phrases like "sentient" (individuals) or "dead bodies", to me, reads more as the populistic approach; I believe it is not necessary.
Other comments are given in the pdf of the reviewed manuscript.

Author Response
Reviewer 1.
Thank you for your valuable and helpful comments.
I revised the manuscript according to the reviewers' and editor’ssuggestions: 1. I included a Simple Summary at the beginning (l. 5); 2. Back Matter section has been added to the manuscript (l. 420) 3. I have added Figure 1 (l. 565) to the manuscript, so the previous Fig. 1 became Fig. 2 (l. 571).
Specific comments:
- In my opinion, the term "sentient" used for at least some aquatic invertebrates is fully justified in accordance with the scientific definition of this term (e.g. Baracchi, Baciadonna. (2020). Insect sentience and the rise of a new inclusive ethics; Powell, Mikhalevich, (2021). Affective sentience and moral protection). Deciding which invertebrates are certainly NOT sentient is difficult, but for at least some insects and crustaceans, using the term is certainly not populism. I am sure that an animal is considered (or not) "sentient" based on arguments from hard, experimental science, not on the basis of personal opinions, worldviews or ideologies. I have therefore decided not to replace this adjective with another.
- I used the term "dead bodies" (l. 339) only to reinforce the argument - perhaps unnecessarily. It has been deleted.
Paweł Koperski
Reviewer 2 Report
The manuscript entitled "It is not only data. Freshwater invertebrates misused in biological monitoring." provides a comprehensive overview of aquatic invertebrate sampling methods and problems. The viewpoints were interesting and of value, suggesting that the mortality of animals used in studies has been shown as excessive. The manuscript is well written and I learnt a lot rather than make too many comments. Despite this, some parts of the manuscript need to be improved. Hope the below comments will be able to help to further improve this work.
Title
I really like the current title.
Introduction
Lines 183-221 These two paragraphs give some data to prove that large numbers of invertebrates are captured and killed, but invertebrate populations are already very large, especially in the case of species that reproduce rapidly, and for which such catches may not represent a significant proportion of their total numbers. It would be possible to compare catches and totals of rare species to better visualise the severity of the impacts of anthropogenic disturbances.
Discussion
Line 231 I can't find where Figure 1 is.
Conclusions
Line 305 I would suggest that the author give the full name of eDNA on its first appearance, instead of it appearing at line 380.
Future directions
Lines 380-408 However, this method is still costly at present and the effect of different PCR primers on the results is significant.
Wouldn't it be more appropriate to put the “Conclusions” after the “Future directions”?
Author Response
Thank you for your valuable and helpful comments. I revised the manuscript according to the reviewers' and editor’ssuggestions: 1. I included a Simple Summary at the beginning (l. 5); 2. Back Matter section has been added to the manuscript (l. 420) 3. I have added Figure 1 (l. 565) to the manuscript, so the previous Fig. 1 became Fig. 2 (l. 571).
Specific comments:
Introduction
Lines 183-221 These two paragraphs give some data to prove that large numbers of invertebrates are captured and killed, but invertebrate populations are already very large, especially in the case of species that reproduce rapidly, and for which such catches may not represent a significant proportion of their total numbers. It would be possible to compare catches and totals of rare species to better visualise the severity of the impacts of anthropogenic disturbances.
Answer: I fully agree with this remark. It is very important and needs to be established, unfortunately there is no (at least I do not know) such data in relation to aquatic invertebrates. I plan to do research on this in the near future. For now, we have to rely on suppositions. Rare and common species cannot be obtained from bio-monitoring data because animals are identified to the family level only and then usually discarded (I mentioned this in l. 203).
Discussion
Line 231 I can't find where Figure 1 is.
Answer: Both figures are put at the end.
Conclusions
Line 305 I would suggest that the author give the full name of eDNA on its first appearance, instead of it appearing at line 380.
Answer: Yes, of course. It is my fault. It has been corrected.
Future directions
Lines 380-408 However, this method is still costly at present and the effect of different PCR primers on the results is significant.
Answer: I agree – the costs are still high, and the efficiency of determining species of freshwater invertebrates with e-DNA methods, especially in the quantitative aspect, is still much lower than with traditional methods based on morphological features. But at the family level, things are, I suppose, not so clear. It seems that the costs of these studies are successively decreasing, and as a result of the world-wide dissemination of these methods, the efficiency of taxa detection using PCR primers is gradually increasing. (Interesting conclusions on this subject can be found in: Fu, Hemery, Sather, N. K. 2021. Cost efficiency of environmental DNA as compared to conventional methods for biodiversity monitoring purposes at marine energy sites - www.pnnl.gov) Therefore, it seems they should be regarded as the future standard of research.
Wouldn't it be more appropriate to put the “Conclusions” after the “Future directions”?
Answer: I basically agree with You, but I don't know if I can do that in accordance with the guidelines for authors. I sent the question to the Editor.
Reviewer 3 Report
A brief summary
The manuscript idea is good, but it lacks evidence to this misuse or mortality for freshwater invertebrates specially in Poland as you mentioned. Also, you need to mention the procedure of biomonitoring to explain for reader how it works so clarify more about the misuse of invertebrates.
Specific comments
Abstract
- Line 7-8: write (using) not (use), remove (of) before (free)
- Line 9-10: write (the) before (ethical), write (the) before (procedures), remove (larger) before (number)
- Line 11-12: write (for) not (of), write (that) before (results), write (such) not (this), write (to be) not (as)
- Line 13: write (was due to) before (unnecessarily), remove (as a results of), remove (the) before (subsampling)
- Line 14: write (which results in) before (excessive), remove (through)
- Line 15-16: not clear meaning, clarify
- Line 17: write (main) not (big)
- Line 19-21: irrelevant
Introduction
- Line 31: write (of) not (in), add (detecting) before (ongoing)
- Line 37: remove (or number or), write (of) before (relative)
- Line 40: write (depends on) not (is to), write (collection) not (collect), write (of) before (whole)
- Line 50-52: repeatable meaning, remove
- Line 53: remove (however)
- Line 63: write (the) before (most)
- Line 69: write (however the) before (less)
- Line 73: remove (freshwater) before (species)
- Line 75-79: not clear, clarify
- Line 121:put in separate heading
- Line 123: FWD or WFD??
- Line 134-135: not clear, clarify
- Line 164: FWD or WFD??
- Line 176-182: not clear, clarify
- Line 189-193: too long, clarify
- Line 194-195: write (their) not (there), write (although) before (24,577)
- Line 230: no figure present
- Line 231-233: not clear, clarify
Conclusion
- Line 297-298: not clear, clarify
Minor editing of English language required.
Author Response
Thank you for all your valuable and helpful comments. I revised the manuscript according to the reviewers' and editor’ssuggestions: 1. I included a Simple Summary at the beginning (l. 5); 2. Back Matter section has been added to the manuscript (l. 420) 3. I have added Figure 1 (l. 565) to the manuscript, so the previous Fig. 1 became Fig. 2 (l. 571).
A brief summary
The manuscript idea is good, but it lacks evidence to this misuse or mortality for freshwater invertebrates specially in Poland as you mentioned. Also, you need to mention the procedure of biomonitoring to explain for reader how it works so clarify more about the misuse of invertebrates.
Answer: The detailed description of the procedure is long, boring and complicated, I thought it would not be attractive to the reader - I added a link to the online version (151). The abridged version can now be found in verses 141-150 and additional important points 180-199. Considerations regarding the recognition of activity as misuse are contained in verses 117-138.
Specific comments (my answers bolded):
Abstract
- Line 7-8: write (using) not (use), remove (of) before (free) – it has been changed
- Line 9-10: write (the) before (ethical), write (the) before (procedures), remove (larger) before (number) – it has been changed
- Line 11-12: write (for) not (of), write (that) before (results), write (such) not (this), write (to be) not (as) – it has been changed
- Line 13: write (was due to) before (unnecessarily), remove (as a results of), remove (the) before (subsampling) – it has been changed
- Line 14: write (which results in) before (excessive), remove (through) – it has been changed
- Line 15-16: not clear meaning, clarify (a link to the relevant section of the article has been added, l. 16)
- Line 17: write (main) not (big) – it has been changed
- Line 19-21: irrelevant - removed
Introduction
- Line 31: write (of) not (in), add (detecting) before (ongoing) it has been changed
- Line 37: remove (or number or), write (of) before (relative) – no, I can’t agree - it is important. In certain methods presence/absence of species is noted in other ones their relative abundance.
- Line 40: write (depends on) not (is to), write (collection) not (collect), write (of) before (whole) it has been changed
- Line 50-52: repeatable meaning, remove it has been changed
- Line 53: remove (however) it has been changed
- Line 63: write (the) before (most) it has been changed
- Line 69: write (however the) before (less) it has been changed
- Line 73: remove (freshwater) before (species) it has been changed
- Line 75-79: not clear, clarify – this fragment has been seriously rearranged.
- Line 121:put in separate heading - ? it is in separate heading?
- Line 123: FWD or WFD?? – WFD of course, ita has been improved
- Line 134-135: not clear, clarify – it has been changed
- Line 164: FWD or WFD?? - WFD of course, ita has been improved
- Line 176-182: not clear - clarify – it has been presented in more, I hope, clear way (l. 186)
- Line 189-193: too long, clarify – it has been a bit shortened
- Line 194-195: write (their) not (there), write (although) before (24,577) – it has been changed
- Line 230: no figure present – both figures are place dat the end
- Line 231-233: not clear, clarify - I added one sentence (l. 243) - maybe it's clearer now?
Conclusion
- Line 297-298: not clear, clarify - I hope the excerpt is clear with the Figure 2.
Paweł Koperski
Reviewer 4 Report
See attached file.

Author Response
Thanks for Your opinion.
I revised the manuscript according to the reviewers' and editor’ssuggestions: 1. I included a Simple Summary at the beginning (l. 5); 2. Back Matter section has been added to the manuscript (l. 420) 3. I have added Figure 1 (l. 565) to the manuscript, so the previous Fig. 1 became Fig. 2 (l. 571).
Paweł Koperski